# Characterizing the Complete Mitochondrial Genomes of Three Bugs (Hemiptera: Heteroptera) Harming Bamboo

**DOI:** 10.3390/genes14020342

**Published:** 2023-01-28

**Authors:** Wenli Zhu, Lin Yang, Jiankun Long, Zhimin Chang, Nian Gong, Yinlin Mu, Shasha Lv, Xiangsheng Chen

**Affiliations:** 1Institute of Entomology, Guizhou University, Guiyang 550025, China; 2The Provincial Special Key Laboratory for Development and Utilization of Insect Resources, Guizhou University, Guiyang 550025, China; 3The Provincial Key Laboratory for Agricultural Pest Management of Mountainous Regions, Guizhou University, Guiyang 550025, China; 4Engineering Research Center of Medicinal Resources and Health Care Products, Guiyang Healthcare Vocational University, Guiyang 550081, China

**Keywords:** bamboo pests, Coreoidae, Lygaeoidae, mitogenome, hazard condition, life history

## Abstract

Herein, we report the mitochondrial genomic characteristics of three insect pests, *Notobitus meleagris*, *Macropes harringtonae,* and *Homoeocerus bipunctatus*, collected from bamboo plants in Guizhou Province, China. For the first time, the damaged conditions and life histories of *M. harringtonae* and *H. bipunctatus* are described in detail and digital photographs of all their life stages are provided. Simultaneously, the mitochondrial genome sequences of three bamboo pests were sequenced and analyzed. *Idiocerus laurifoliae* and *Nilaparvata lugens* were used as outgroups, and the phylogenetic trees were constructed. The mitochondrial genomes of the three bamboo pests contained 37 classical genes, including 13 protein-coding genes (PCGs), two ribosomal RNA genes (rRNAs), 22 transfer RNAs (tRNAs), and a control region, with a total length of 16,199 bp, 15,314 bp, and 16,706 bp, respectively. The A+T values of the three bamboo pests were similar, and trnS1 was a cloverleaf structure with missing arms. The phylogenetic analyses, using the Bayesian inference (BI) and Maximum likelihood (ML), supported that *N. meleagris* and *H. bipunctatus* belonged to the Coreoidea family, whereas *M. harringtonae* belonged to the Lygaeoidea family with high support values. This study involves the first complete sequencing of the mitochondrial genomes of two bamboo pests. By adding these newly sequenced mitochondrial genome data and detailed descriptions of life histories, the database of bamboo pests is improved. These data also provide information for the development of bamboo pest control methods by quick identification techniques and the use of detailed photographs.

## 1. Introduction

Bamboo is a common plant that belongs to the Gramineae family. Bamboo resources are an essential part of the terrestrial forest ecosystem with typical characteristics of wide distribution, rapid growth, high yield, strong regeneration ability, wide use, and high economic value. Bamboo has high economic, ecological, and social benefits, and is widely used in the construction, ornamental, and food industries. Additionally, the bamboo extracts exhibited excellent anti-free radicals, antioxidant, anti-aging, antibacterial, insecticidal, lipid regulation, cardiovascular and cerebrovascular protection, and pharmacological effects [1]. There are more than 120 genera and over 1500 species of bamboo worldwide, and approximately 44 genera and 762 species of bamboo in China [2]. With increased bamboo planting areas, bamboo pests become more common, and damage becomes more severe, which hinders the sustainable development of the bamboo industry. While investigating bamboo pests in Guizhou, the authors found three species that are serious bamboo pests: *N. meleagris*, *M. harringtonae*, and *H. bipunctatus*, which belong to the Hemiptera (Heteroptera) order [3,4,5]. At the end of the 19th century and the beginning of the 20th century, Japanese researchers studied the behavior of *N. meleagris*. They noticed that *N. meleagris* have the habit of multi-male mating aggregation [6,7]. Many Chinese researchers have also reported the habits of *N. meleagris*. Adults and nymphs of *N. meleagris* are harmful because they suck sap through thorns. Bamboo forests were severely damaged by these pests, which killed 70% of the bamboo plants. The harmed bamboo withered, bamboo whips rotted, and dried without osmotic fluid. This led to the following year’s reduction of bamboo shoots and the decline of the forest [8]. According to our study, the damage of *N. meleagris* in Guizhou is becoming increasingly severe, and there are over20 insects on each bamboo shoot simultaneously. No scholars have studied the biological characteristics of *M. harringtonae* and *H. bipunctatus* before. This study is the first to report these insects’ effects on bamboo; they suck sugar and other nutrients from the bamboo rod’s basal membrane and inner wall with their piercing mouthparts, and the damaged sites often become reddish-brown or black. The bottom of the node becomes corroded and withered. *M. harringtonae* can cause damage to every type of bamboo. Therefore, the damage and life histories of *M. harringtonae* and *H. bipunctatus* are described in this study.

Mitochondrial genome sequences are widely used in biogeographical, molecular, and systematic studies [9,10]. Mitochondrial genome research includes explaining species’ origins and exploring insects’ phylogeny, revealing the geographical distribution of species polymorphisms. This relationship provides several genome level characteristics, including changes in genomic diversity, control patterns for transcription and replication, and RNA secondary structures (such as cloverleaf structures) [11,12]. Because of the higher base replacement rate than nuclear genes, due to the lack of rearrangement during cell meiosis, these characteristics make mitochondrial DNA a focal genetic marker for evolutionary studies [13,14,15,16]. So far, only a few mitochondrial bamboo pests (such as *Notobitus montanus*, *Pirkimerus japonicus*, *Hippotiscus dorsalis,* and *Yemmalysus parallelus*) have been sequenced and are available on NCBI [17,18,19]. There were no complete mitochondrial sequences of *N. meleagris* and *H. bipunctatus* on NCBI. This study presents the complete mitochondrial genome of three bamboo pests (*N. meleagris*, *M. harringtonae*, and *H. bipunctatus*), which provides the basis for developing bamboo-pest gene bank data and supports prevention and management. This is the first study to investigate bamboo pests by using the mitochondrial genomes data and life histories with detailed molecular and morphological datasets. We also discuss their mitochondrial genome structures and analyze their tRNA’s shamrock structure. This study aims to provide a reference for the identification, control, and phylogenetic analysis of bamboo pests.

## 2. Materials and Methods

### 2.1. Observation of the Damaged Condition and Occurrence Regularity

From May 2021 to August 2022, we observed the damaged conditions and occurrence regularity of three bamboo pests (*N. meleagris*, *M. harringtonae*, and *H. bipunctatus*) in Guizhou Province. The observations were madeon sunny days, at an interval of once every ten days. The damages were recorded and photographed using visual inspection and sweep net techniques [20].

### 2.2. Sample Isolation and DNA Extraction

*N. meleagris* was collected in July 2021 from the Baizi Bridge, Duyun, and Guizhou, whereas *M. harringtonae* and *H. bipunctatus* were collected in August 2021 from Huaxi Park, Guiyang, and Sajinriver, Fuquan, Guizhou, respectively. Identification was made based on external body morphology and genitalia with the help of theavailable literature [21,22,23]. After 48 h of starvation, fresh individuals were preserved in 95% ethanol at −40 °C at the Guizhou Provincial Key Laboratory for Agricultural Pest Management of the Mountainous Regions, Guizhou University. Total DNA was extracted from the entire body using the Genomic Extraction Kit [24,25,26,27].

### 2.3. Genome Assembly, Annotation, and Analysis

The DNA quality before sequencing was evaluated using agarose (1%) gel electrophoresis. Mitogenomes were sequenced using a next-generation sequencing platform with Illumina Hiseq 2500 at BerryGenomics (Beijing, China). BerryGenomics (Beijing, China) and FastQC (www.bioinformatics.babraham.ac.uk/projects/fastqc accessed on 17 March 2022) were used to evaluate the quality of the raw sequences. Then, the clean sequences were assembled using MitoZ v2.4 software [28] with default parameters and the mitogenomes of *Riptortus pedestris* (Alydidaeidae; NC_012462), *Aeschyntelus notatus* (Coreidae; NC_012446), and *Geocoris pallidipennis* (Geocoridae; NC_012424) were used as references. The three mitogenomes were initially annotated using the MITOS web server (http://mitos.bioinf.uni-leipzig.de/index.py, accessed on 1 January 2023) [29] with invertebrate genetic codes. Using the MITOS web server, we identified and predicted 22 tRNA genes’ locations and secondary structures. The 13 protein-coding genes (PCGs) were predicted by determining their open reading frames using the invertebrate mitochondrial genetic codons [30,31,32]. The skews of AT and GC were calculated according to the following formulas: AT skew = (A − T)/(A + T) and GC skew = (G − C)/(G + C) [29,33,34]. The nucleotide composition and relative synonymous codon usage (RSCU) were obtained using PhyloSuite v1.2.2 [35], and RSCU figures were created using the package [36] of R 3.6.1 [37].

### 2.4. Phylogenetic Analysis

In addition to the three mitogenomes sequenced in this study, phylogenetic analyses were conducted based on an additional 22 complete mitogenomes of the Hemiptera species from NCBI. The Hemiptera species belonged to 6 superfamilies: Lygaeoidea (11 species), Coreoidea (8 species), Pentatomoidea (2 species), Reduvioidea (2 species), Fulgoroidea (1 species), and Membracoidea (1 species). The mitogenomes of *I. laurifoliae* (Cicadellidae) and *N. lugens* (Delphacidae) from Auchenorrhyncha were selected as the outgroup (Table A1). Accession numbers and detailed information on these mitogenomes are listed in Table A1. We used MEGA v6 [38] to align the nucleotide sequences of 13 PCGs with Muscle [39,40], and used SequenceMatrix v1.7 [41] to concatenate individual genes. Model testing and selection was completed by using the software PartitionFinder v2.1.1 [42] with the greedy algorithm [43]. Maximum likelihood (ML) analyses were employed using IQ-TREE v1.6.3 [44] with 10,000 replicates of ultrafast likelihood bootstrapping [45]. Bayesian inference (BI) analyses were employed using MrBayes 3.2 [46] under the matrix of two simultaneous operations of 1,000,000 generations, sampling every 1000 generations, with a burn-in of 25%. When the splitting frequency drops steadily to 0.01, the sample is considered to have converged. Finally, using FigTree v.1.4.3 [47], we viewed and beautified the resulting phylogenetic trees [48,49].

## 3. Results

### 3.1. Hazard Condition and Occurrence Regularity

According to our investigation of the three bamboo pests in Guizhou Province from 2021 to 2022, the preliminary observation showed that *N. meleagris* was significantly harmful to bamboo shoots, mainly harming them with clusters; *M. harringtonae* can harm whole bamboo tree, including bamboo poles and bamboo joints, and have a wide distribution range; *H. bipunctatus* often inhabits the growth of newly emerging leaves and feed on them. Furthermore, the insect pests of bamboo species often aggregate and harm the bamboo plants. The adults and nymphs are also harmful as they suck sap through the stylets (Figure 1, Figure 2 and Figure 3).

*N. meleagris*, five instars, mass egg production with at least 20 eggs per time, obvious generation overlap, >3 generations a year, and adults overwinter in dry trees; *M. harringtonae*, five instars, mass egg production with at least 20 eggs laid per time, >3 generations a year, overwintering as adults in bamboo nodes; *H. bipunctatus*, five instars, single egg production, >2 generations a year, and adults overwintering in weeds. (Table A2).

### 3.2. Mitogenomic Organization and Composition

The mitogenomes of the three bamboo pests, *N. meleagris* (GenBank No. OP442510; length: 16,199 bp), *M. harringtonae* (GenBank No. OP442511; length: 15,314 bp), and *H. bipunctatus* (GenBank No. OP442512; length: 16,706 bp) were double-stranded closed circular molecules (Figure 4). The newly sequenced mitogenomes of three bamboo pests presented 37 typical metazoan mitochondrial genes. These were similar to the mitogenomic sequences of other Hemipteran insects [50,51,52,53], containing 13 PCGs, 22 tRNA genes, two rRNA genes, and a control region (Table A3). Each sequence of the three bamboo pests included nine PCGs and 14 tRNAs encoded on the major (J-strand), and the minor (N-strand) consisted of four PCGs, eight tRNA, and two rRNAs. In addition, there were some differences between the overlapping regions and intergenic spacers of the three mitogenomes. There were seven overlapping regions and 12 intergenic spacers of *N. meleagris*, the largest overlapping region was 7 bp located between atp8 and atp6, and the largest intergenic spacer was 37 bp located between trnY and cox1. In addition, there were 11 overlapping regions and 10 intergenic spacers in *M. harringtonae*; the largest overlapping region was 7 bp between atp8 and atp6, and the largest intergenic spacer was 71 bp between trnH and nad4. There were nine overlapping regions and 13 intergenic spacers in *H. bipunctatus*; the largest overlapping region was 8 bp between trnW and trnC, and the largest intergenic spacer was 37 bp located between trnY and cox1. The nucleotide compositions of *N. meleagris, M. harringtonae,* and *H. bipunctatus* are shown in Table A4. The AT nucleotide content of the three mitogenomes was similar: In the range of 73–74.5%, the content occupoed a substantial proportion of the entire sequence. The AT skew of all three genomes is a positive number; on the contrary, the GC skew of all three genomes is a negative number.

### 3.3. PCGs and Codon Usage

The mitogenomes of the three bamboo pests belong to the Hemipteran order [47], which includes 13 PCGs. Their lengths in *N. meleagris*, *M. harringtonae*, and *H. bipunctatus* were 11,008 bp, 10,957 bp, and 11,010 bp, respectively. In the three sequences, the nine PCG genes (nad2, cox1, cox2, atp8, atp6, cox3, nad3, nad6, and cytb) were encoded on the major strand (J-strand), and four PCG genes (nad5, nad4, nad4L, and nad1) were encoded on the minor strand (N-strand). All 13 PCGs started with ATN. The stop codon of *N. meleagris* is the same as that of *H. bipunctatus*, atp8 and nad6 had TAA as the stop codon, and the other ten had incomplete T. The stop codon of *M. harringtonae* is special, except for the same features as the other two sequences, nad4L had TAA as the stop codon, and nad3 had TAG as the stop codon.

Except for the stop codon, the total number of codons was 3663 (*N. meleagris*), 3645 (*M. harringtonae*), and 3664 (*H. bipunctatus*). In descending order, the three most abundant amino acids, Leu2, Ile, and Phe, in *N. meleagris* are the same as *M. harringtonae*. In addition, Leu2, Ile, and Met were the most abundant amino acids in *H. bipunctatus* (Figure 5). According to Figure 6, the four most prevalent codons were Leu2 (UUA), Ile (AUU), Phe (UUU), and Met (AUA). The RSCU values of the PCGs indicated a pattern toward more A and T than G and C.

### 3.4. Transfer and Ribosomal RNA Genes

The rrnL (16S) and rrnS (12S) genes on the N-strand were located between the trnL1 and trnV and the control region in the mitogenome of three bamboo pests (Table A3). The total lengths of rrnL and rrnS of the three sequences were similar, in the range of 2036 bp to 2067 bp, and displayed a negative AT skew and a positive GC skew (Table A4).

The mitogenomes of *N. meleagris*, *M. harringtonae*, and *H. bipunctatus* included 22 transfer RNA genes, as in most invertebrates. The total lengths of tRNAs were 1449 bp, 1439 bp, and 1446 bp, and these tRNA genes ranged from 62–75 bp. In the three sequences, the 14 tRNA genes (trnI, trnM, trnW, trnL2, trnK, trnD, trnG, trnA, trnR, trnN, trnS1, trnE, trnT, and trnS2) were encoded on the major strand (J-strand), and the eight tRNA genes (trnQ, trnC, trnY, trnF, trnH, trnP, trnL1, trnV) were encoded on the minor strand (N-strand) (Table A3). We found that only trnS1 lacked the dihydrouridine (DHU) arm, and the remaining 21 tRNA genes can form a typical cloverleaf structure (Figure A1, Figure A2 and Figure A3). In addition to the typical base pairing (G-C and A-U), there was some wobble G-U pairs in these secondary structures, which could form stable chemical bonds between U and G.

### 3.5. Control Region

The control region, also called the A+T rich region, is the longest noncoding region with many genes involved in mitogenic replication and transcription. In the three bamboo pests, this region was located between the rrnS and trnL. The length of the control region was 1627 bp (*N. meleagris*), 772 bp (*M. harringtonae*), and 2138 bp (*H. bipunctatus*). The AT-rich region had the highest AT content with67.2% in *N. meleagris*, 79.3% in *M. harringtonae*, and 68.4% in *H. bipunctatus*, with positive AT skew (0.099–0.117) and negative CG skew (−0.329 to −0.188) (Table A4).

### 3.6. Phylogenetic Analyses

Phylogenetic relationships among 23 species of the heteropteran (including the three sequenced mitogenomes of the bamboo pests, two of them newly sequenced) and two outgroups (*I. laurifoliae* and *N. lugens*) were reconstructed based on 13 PCGs using ML and BI analyses under the partitioning scheme and models selected by PartitionFinder. The two resulting trees (Figure 7) had similar topologies, receiving strong support in most nodes. These phylogenetic relationships were consistent with previous studies [54,55]. The phylogenetic trees of *N. meleagris* and *H. bipunctatus* from sister group relationships, *M. harringtonae* with *M. robustus*, and *M. dentipes* also from sister group relationships, showed a high confidence value. The sister groups’ relationship of Coreoidea and Lygaeoidea located in the middle of phylogenetic trees was also confirmed [56,57,58].

## 4. Discussion

*N. meleagris* are typical insects that attack and harm bamboo plants [59]. *M. harringtonae,* belonging to the family Blissidae, and many genera (*Macropes* and *Pirkimerus*), have been reported to have harmed bamboo, but this is the first time that *M. harringtonae* has been reported to have harmed bamboo seriously [60,61]. *Homoeocerus*, belonging to the Coreidae family, have previously been reported to harm only leguminous plants [62]; however, *H. bipunctatus* collected from the bamboo plants showed normal physiological activities, such as mating and oviposition, when the species were fed with bamboo. This study describes the extent of damages and life histories of *M. harringtonae* and *H. bipunctatus*. This paper updates and supplements the data on *N. meleagris*, *M. harringtonae*, and *H. bipunctatus* in Guizhou. Research in 2009 showed that only two generations of *N. meleagris* were present in Guizhou in one year [63]. However, according to our research observations, at least three generations of *N. meleagris* were present in Guizhou in one year, and the generations overlapped significantly. Previously, no study was performed on the biological characteristics of *M. harringtonae* and *H. bipunctatus.* Our research shows that at least three generations of *M. harringtonae* occur in a year in Guizhou, and at least two generations of *H. bipunctatus* occur in a year in Guizhou. This study fills a gap in the understanding of the biological characteristics of *M. harringtonae* and *H. bipunctatus*.

In this study, three complete mitogenomes (*N. meleagris*, *M. harringtonae*, and *H. bipunctatus*) were sequenced and analyzed for their genome size, base content, AT nucleotide bias, AT skew, GC skew, the codon usage of protein genes, and secondary structure of tRNA. Despite differences in the sequence length of three bamboo pest species, the mitochondrial genome order of *N. meleagris*, *M. harringtonae*, and *H. bipunctatus* was identical and conserved with the alignment to that of known ancestral taxa regarding the organization and composition of the genome [64,65,66,67,68]. The size, AT skew, and GC skew of genome and PCGs of *M. harringtonae* in our study has little difference with previous research. However, the positive and negative of AT skew and GC skew of rRNAs and tRNAs are opposite [17]. The analysis of synonymous codon usage showed that the occurrence of synonymous codons ending in A or T was much higher than those of other synonymous codons; that is, codons rich in AT were frequently used. Whether the abundant AT content in the control region affects transcription and replication of the mitogenome and indirectly affects the feeding behavior of the bamboo pests is unclear; further studies are required to verify the function of the conserved control region of the mitogenome of the bamboo pests. In the insect mitochondrial genome, the stem-loop structure with dihydrouracil deletion of trnS1 in the tRNA secondary structure is a typical feature [69,70,71]. The trnS1 secondary structures of the three mitochondrial genome sequences of bamboo pests are stem-loop structures with dihydrouracil deletion; the other 21 tRNA secondary structures were typical clover structures.

The phylogenetic tree was constructed using 13 protein genes, and the phylogenetic relationships of 25 species were analyzed. The results showed that *N. meleagris* and *H. bipunctatus* belonged to the Coreoidea, and *M. harringtonae* belonged to the Lygaeoidea. There was no dispute on the taxonomic status, consistent with the results of morphological identification [17,72,73]. The three mitogenomes sequenced enriched the database of Heteroptera and laid a foundation for better resolving the controversy of the taxonomic status of bugs. The study of the predator web of natural enemy insects is to determine the prey of natural enemy insects by measuring the DNA fragments of the intestinal contents of natural enemy insects and comparing them in the database [74]. However, before that, it is necessary to establish a database of insect pests and surrounding arthropod species. Therefore, this study will also provide essential information for subsequent research on analyzing predator nets of natural enemies of bamboo pests.

## Figures and Tables

**Figure 1 genes-14-00342-f001:**
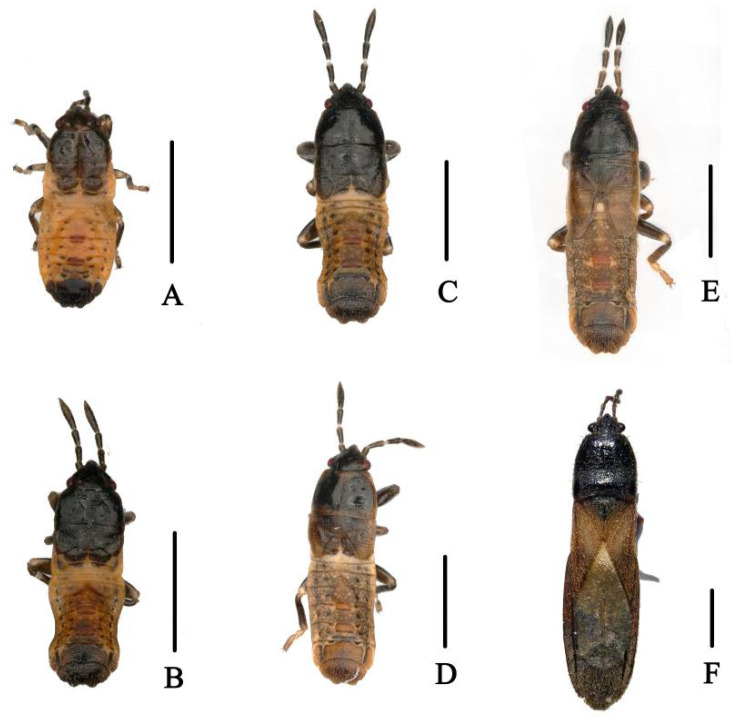
Dorsal habitus of *M. harringtonae*. (**A**). First instar; (**B**). Second instar; (**C**). Third instar; (**D**). Fourth instar; (**E**). Fifth instar; (**F**). Adult. Scale bar = 1000 μm.

**Figure 2 genes-14-00342-f002:**
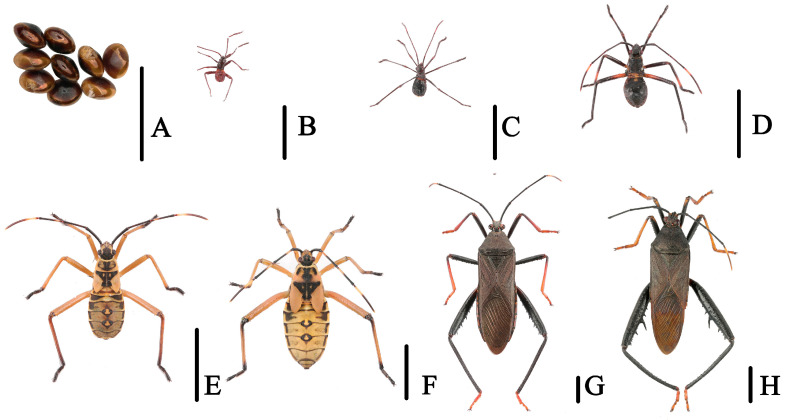
(**A**). Egg mass of *N. meleagris*; (**B**–**F**). Dorsal habitus of first to fifth nymphal instars; (**G**). Adult, female; (**H**). Adult, male. Scale bars = (**A**) (500 μm); (**B**–**F**) (5000 μm).

**Figure 3 genes-14-00342-f003:**
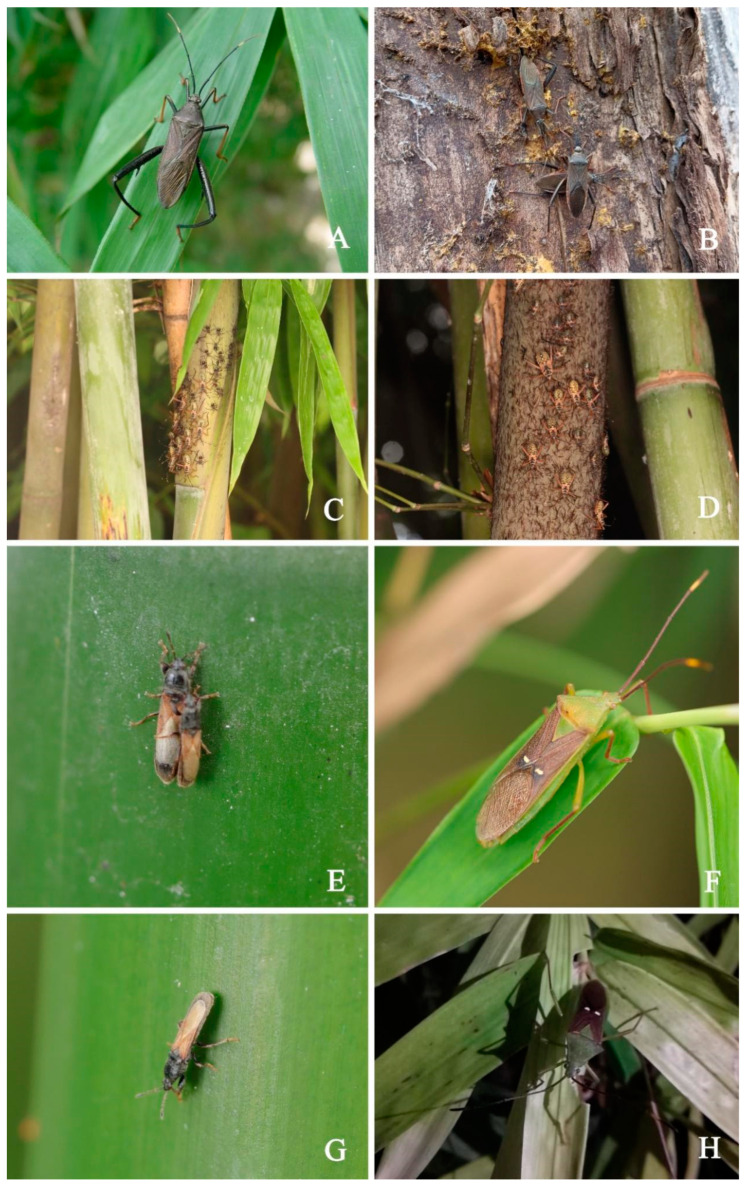
(**A**,**B**). Living dorsal habitus of adult *N. melegaris*; (**C**,**D**). Nymphal stages of *N. melegaris*; (**E**,**G**). *M. harringtonae*, adults; (**F**,**H**). *H. bipunctatus*, adults.

**Figure 4 genes-14-00342-f004:**
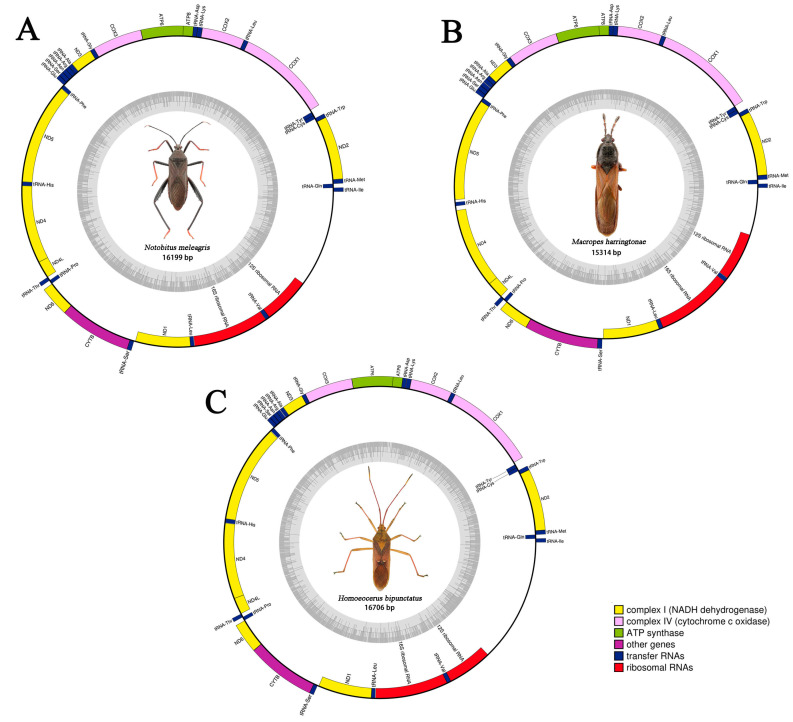
Circular maps of the mitogenomes of *N. meleagris* (**A**), *M. harringtonae* (**B**), and *H. bipunctatus* (**C**). The pink, green, yellow, and purple show PCGs, blue shows tRNAs, red shows the rRNAs, and blank shows the control region. Insect morphology and sequence length are indicated in the center of the circle diagram.

**Figure 5 genes-14-00342-f005:**
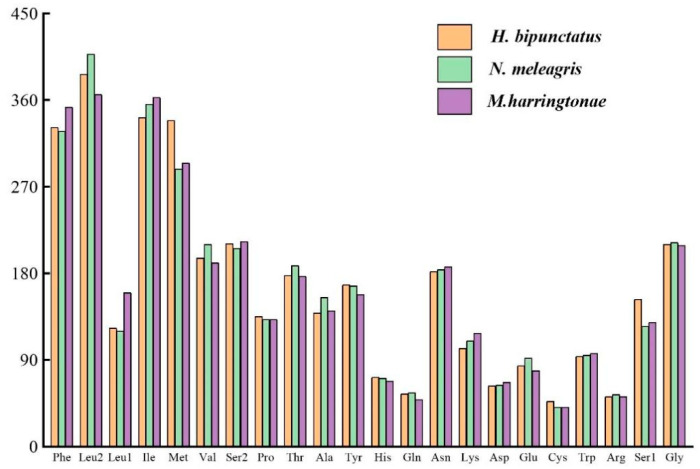
Codon distribution in three bamboo pest species: the color-filled orange blocks indicate *H. bipunctatus*, the filled green blocks represent *N. meleagris*, and the filled violet blocks show *M. harringtonae*. The total number of the codons are presented as numbers at the *Y*-axis and codon families are shown at the *X*-axis.

**Figure 6 genes-14-00342-f006:**
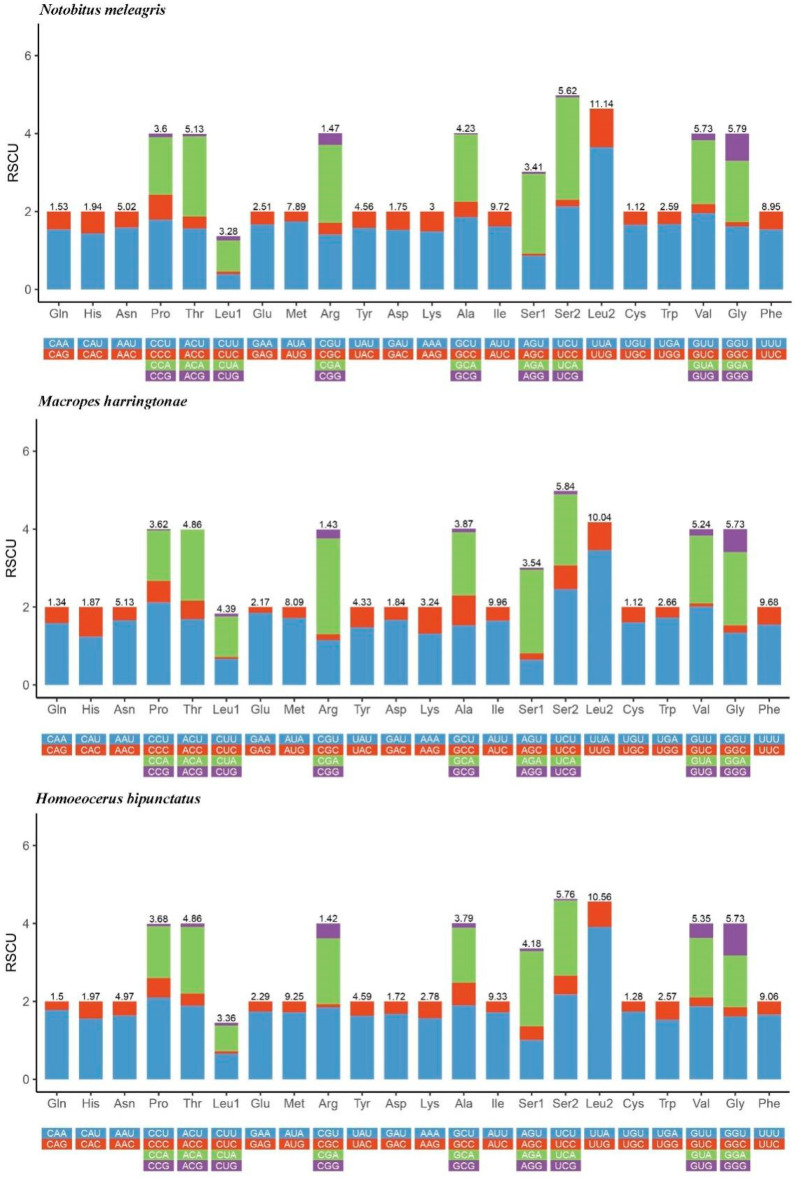
Relative synonymous codon usage (RSCU) within *N. meleagris*, *M. harringtonae* and *H. bipunctatus*. Codon families are shown on the *X*-axis and the frequency of RSCU on the *Y*-axis.

**Figure 7 genes-14-00342-f007:**
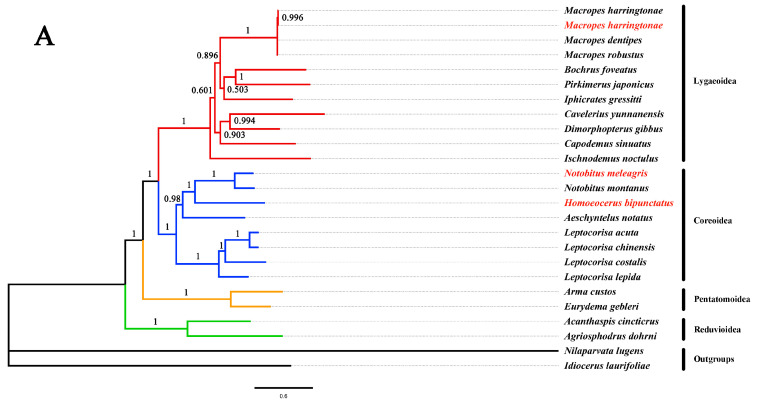
Phylogenetic relationships based on Bayesian inference (**A**) and maximum likelihood (**B**) analyses. Bayesian posterior probabilities are shown on each node. Bootstrap support values are shown on each node. The tree was rooted using *I. laurifoliae* and *N. lugens* as outgroups. The three bamboo pest species sequenced in this study are marked in red.

## Data Availability

The *N. meleagrare*, *M. harringtonae*, and *H. bipunctatus* mitogenome sequence was submitted to NCBI (Acc. number).

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
