# Peer review of "Characterizing the Complete Mitochondrial Genomes of Three Bugs (Hemiptera: Heteroptera) Harming Bamboo"

_genes, 2023, doi:10.3390/genes14020342_

Round 1

Reviewer 1 Report (Previous Reviewer 2)

The accession numbers of the three mitogenomes are missing and not included in the section of the data Availability Statement. The results cannot be assessed without having access to the mitochondrial sequences and their full annotations. Please email the NCBI team to release the data.

Author Response

Reviewer 2 Report (New Reviewer)

I have thoroughly revised this manuscript and several minor suggestions and modifications are directly implemented in attached pdf file. Before formal acceptance, the following recommendations and queries must be answered as attached in the word file and pdf. 

Author Response

Reviewer 3 Report (New Reviewer)

I suggest the modifications below:

Line 3

Change “(Hemiptera:Heteroptera)” for “(Hemiptera: Heteroptera)”

Line 14

Change “meleagrare” for “meleagris”

Line 28

Change “Heteroptera” for “Coreidae; Lygaeidae”

Lines 72-74

“So far, only few mitochondrial genome bamboo pests (such as Notobitus montanus, Pirkimerus japonicus, Hippotiscus dorsalis and Yemmalysus parallelus) have been sequenced in NCBI.” - Authors need to cite papers describing the mitochondrial genomes of these species.

Line 77

Change “H. Bipunctatus” for “H. bipunctatus”

Line 257

Change “N. meleagris” for “Notobitus meleagris” - At the beginning of the sentence you must put the gender in full.

Line 258

Change “M. harringtonae” for “Macropes harringtonae”

Appendix

Why was the phylogeny presented as an appendix? I suggest that a high resolution version of the phylogeny be made and be included as an image in the manuscript.

Author Response

This manuscript is a resubmission of an earlier submission. The following is a list of the peer review reports and author responses from that submission.

Round 1

Reviewer 1 Report

The manuscript is very well written and can be accepted in its current form

Minor amendments

Add in image scale for Figure 2, 3

Reviewer 2 Report

This is an interesting study. The authors have sequenced, assembled, and annotated the mitochondrial genomes of three insect pests, Notobitus meleagrare, Pirkimerus japonicus, and Homoeocerus bipunctatus collected from bamboo plants and compared them with other mtDNA datasets. They constructed a phylogenetic tree, which indicates that N. meleagris and H. bipunctatus belonged to the Coreidae family, and  P. japonicus belonged to the Lygaeidae family. This work is important and has provided valuable information about evolutionary studies and insights into phylogenetic relationships. However, I would suggest some minor revisions as follows:

1- The accession numbers of three mitochondrial genomes are missing. Please make sure you make the three accession numbers available in Genebank.

2- I suggest adding accession numbers for each species to follow the scientific names in the phylogenetic tree (Figure 4).

3- I would suggest improving the quality of the figures to increase the readability if possible.

Reviewer 3 Report

Dear Editor,

The quality of the paper is not up to the level. So, therefore this reviewer would like to suggest that the manuscript is not acceptable for publication in the journal. 

Reviewer 4 Report

The mitogenome of Pirkimerus japonicas had been published in insects journal in 2022 July 17 (Shujing Wang et al. 2022). So the paper title, first report of complete mitochondrial genomes of three bugs (Hemiptera:Heteroptera) harming bamboo and related presentation in abstract & main text (e.g. Line 70-71) are incorrect.

Line 59-60. “No scholars had studied H. bipunctatus before. Therefore, this study describes the damage and occurrence of H. bipunctatus [9].” These sentences and citation are difficult to understand.

Line 64. : What’s mean about ” This relationship”?

Line 68-70, So far, only two mitochondrial bamboo pests (Hippotiscus dorsalis and Yemmalysus parallelus) have been sequenced in NCBI. This sentence is so absolute and inaccurate.

Line237-240, English expression and writing of these sentences are irregular.

Line246, fed with bamboo.

Line254-255 “Previously, there were no reports on the biological characteristics of H.bipunctatus”,this english expression is inaccurate.

Line279 The three gene sequences should be “The three mitogenomes”

Line281-283 these sentences are hard to understand.

Line286 “predator nets”,this word group is difficult to understand.